# Validation of the Multidimensional Fatigue Inventory with Coronary Artery Disease Patients

**DOI:** 10.3390/ijerph17218003

**Published:** 2020-10-30

**Authors:** Julija Gecaite-Stonciene, Adomas Bunevicius, Julius Burkauskas, Julija Brozaitiene, Julius Neverauskas, Narseta Mickuviene, Nijole Kazukauskiene

**Affiliations:** Laboratory of Behavioral Medicine, Neuroscience Institute, Lithuanian University of Health Sciences, 2100 Palanga, Lithuania; adomas.bunevicius@lsmuni.lt (A.B.); julius.burkauskas@lsmuni.lt (J.B.); julijabrozaitiene@gmail.com (J.B.); julius.neverauskas@lsmuni.lt (J.N.); narseta.mickuviene@lsmuni.lt (N.M.); nijole.kazukauskiene@lsmuni.lt (N.K.)

**Keywords:** coronary artery disease, mental exertion, physical performance, psychometric properties, multidimensional fatigue inventory, fatigue, reliability and validity, rehabilitation

## Abstract

Background: Fatigue is a common distressing symptom in patients with coronary artery disease (CAD). The Multidimensional Fatigue Inventory (MFI) is used for measuring fatigue in various clinical settings. Nevertheless, its multidimensional structure has not been consistent across studies. Thus, we aimed to psychometrically evaluate the MFI in patients with CAD. Methods: In sum, 1162 CAD patients completed questionnaires assessing their subjective fatigue level (MFI-20), mental distress symptoms (HADS, STAI), and health-related quality of life (SF-36). Participants also completed exercise capacity (EC) testing. Results: Confirmatory factor analysis of the four-factor model, showed acceptable fit (CFI = 0.905; GFI = 0.895; NFI = 0.893, RMSEA = 0.077). After eliminating four items, confirmatory factor analysis testing showed improvement in the four-factor model of the MFI-16 (CFI = 0.910; GFI = 0.909; NFI = 0.898, RMSEA = 0.077). Internal consistency values were adequate for the total score and four MFI-16 subscales: General fatigue, physical fatigue, reduced activity, and mental fatigue with Cronbach’s α range: 0.60–0.82. The inadequate value (Cronbach’s α = 0.43) was received for the subscale of reduced motivation in both MFI-20 and MFI-16. Correlations between the MFI-16 and HADS, STAI, SF-36, and EC measures were statistically significant (all *p*’s < 0.001). Conclusions: The Lithuanian version of the modified MFI of 16 items showed good factorial structure and satisfactory psychometric characteristics, except for reduced motivation subscale.

## 1. Introduction

Despite the strides made in medicine, coronary artery disease (CAD) remains the most frequent cause of mortality, accounting for almost one third of all deaths globally [1] and impairing not only the individual’s personal life, but also their career and work [2]. Fatigue, which is characterized as a subjective experience of persistent and extreme exhaustion, a lack of energy, and tiredness [3,4,5] is one of the most frequent and stress-provoking symptoms reported by individuals with heart-related conditions [6,7,8,9]. In CAD patients, the prevalence of moderate to severe fatigue reaches 39% during cardiac rehabilitation (CR), remaining up to 28% after one year [10]. Up to date, mental fatigue is considered as a risk factor for developing heart diseases [11], while unusual fatigue is a strong predictor of a longer prehospital delay [12], poor health related outcomes, and an increased risk for mortality [13], which is also considered as one of the key prodromal factors in those after acute coronary syndrome (ACS) [14]. Fatigue in those with heart diseases is also common and distressing symptom that raises concerns within the field of occupational health [2].

Extensive literature of previously published studies suggests strong associations between fatigue and mental distress [7,8,15,16,17,18,19,20,21] as well as poor health-related quality of life (HRQoL) [9] in patients with CAD. Findings in post-ACS patients showed significant associations between fatigue, anxiety and depressive symptoms [7,15,17,18,19]. A study by Eckhardt et al. [22] reported that symptoms of depression were a sole predictor of fatigue intensity. In fact, some authors suggest that there are great conceptual similarities between the construct of depression and fatigue [17,23,24]. Furthermore, previous scientific literature has also drawn the attention towards fatigue and its effect on HRQoL of patients with various clinical conditions [25,26], including heart related conditions [9]. Staniute and colleagues [9] reported that greater limitations due to emotional and physical issues, decreased vitality, poorer social functioning, the perception of worse general health and lower overall HRQoL were significantly linked to greater levels of fatigue. In another study by Burkauskas et al. [8] it was noted that specifically mental, but not other type of fatigue characteristics, was independently associated with worse cognitive functioning in CAD patients during CR, drawing the attention to consider not only physical fatigue and exercise capacity (EC), but also mental fatigue as a risk factor for worse health related outcomes.

Considering that almost one third of CAD patients who completed CR program [10] remains significantly fatigued, it is important to detect the presence of fatigue and its characteristics in those individuals early and accurately as well as adapt the CR program accordingly. Optimal treatment of those with fatiguing illnesses requires measuring severity, frequency, duration, the nature and type of fatigue, as this may help to individualize and monitor the progress of therapy focused on reducing symptoms of fatigue [5]. To date, there are at least 25 standardized unidimensional and multidimensional scales designed to assess fatigue in various clinical populations [27,28]. Nevertheless, well-validated instruments in those with heart related conditions, including CAD, are still lacking.

Multidimensional Fatigue Inventory (MFI), created by Smets and colleagues in 1995 [5], was originally intended as the 20-item self-report method to evaluate five domains of fatigue in oncology patients, including general, physical, and mental fatigue, as well as reduced activity and motivation. The MFI is considered as one of the four most valid instruments, meeting the quality assurance requirements and was found to be the most appropriate tool for assessing fatigue in patients with cancer [27].

Furthermore, recent studies suggest the MFI as a useful tool to measure the level of fatigue not only in cancer-related but also in other clinical populations, including patients with hepatitis B infection [29], Hodgkin’s lymphoma patients [30], those with a chronic fatigue syndrome [31], idiopathic Parkinson’s disease patients [32], fibromyalgia patients [33], individuals with schizophrenia spectrum disorders [34], a major depressive episode [35], multiple sclerosis [36], acquired brain injury [37], pospoliomyelitis syndrome [38], and overall in those that are critically chronically ill and follow intensive care [39]. There were also some attempts to measure validity and reliability of the MFI in 204 patients after myocardial infarction (MI) [40] in a Swedish population, as well as in 201 CAD patients in Brazil [41], suggesting adequate validity and reliability. However, in a larger and more diverse CAD population, the MFI validation studies are very sparse, especially the ones that explore the MFI factorial structure. Nevertheless, the psychometric studies on various self-reported instruments [42,43] in those with heart related conditions are of great importance, as it helps to evaluate the usefulness of the measures and confidently apply research evidence in a clinical work [44].

It is important to note that previous literature has detected several issues regarding the MFI adaptation. Inadequate reliability of the MFI reduced motivation scale was reported in French-Canadian version [45], Swedish version [46], and English version [47,48] of the MFI. Further, in several articles some complications with factorial structure were addressed: the studies of the American [47] and French-Canadian [45] versions of the MFI reported different item loading from the original MFI version of Smets et al. [5], thus did not support the originally proposed structure solution. In the most recent study by Hinz et al. [49] completed in several countries reported incomplete factorial validity of the MFI and the adaptation of the modified MFI was not suggested. These issues suggest that the MFI could be further improved and analyzed in the adaptation process of Lithuanian MFI version by applying factor analysis.

The goal of our research was to evaluate psychometric characteristics of the MFI, including the dimensional structure, reliability and validity of the MFI as well as to investigate the dimensional structure in a large sample of Lithuanian CAD patients. In this current study we hypothesized that in CAD patients after ACS (1) The internal reliability of the total fatigue score and the four MFI scales (i.e., general fatigue, physical fatigue, reduced activity, and mental fatigue) will be satisfactory (2) the MFI scores will be associated with mental distress (HADS and STAI) and HRQoL (SF-36), suggesting good convergent validity of the MFI.

## 2. Materials and Methods

### 2.1. Study Participants

For this cross-sectional study, a sum of 1287 consecutive patients with CAD were invited to take part as the study participants from February 2010 to March 2020. This study was conducted as part of the larger ongoing research exploring biopsychosocial and environmental risk factors affecting the course and progression of CAD. Inclusion criteria were (1) a current diagnosis of CAD after recent ACS; (2) attendance of an in-patient cardiovascular rehabilitation (CR) program at the Hospital Palangos klinika, Neuroscience Institute, Lithuanian University of Health Sciences, Palanga, Lithuania. The patients were admitted to the CR program within one week following ACS treatment and gave written informed consent. Exclusion criteria were (1) the unstable cardiovascular condition (*n* = 62), (2) a severe comorbid illness (*n* = 33), and (3) unwillingness to participate in the study (*n* = 30). In total, after exclusion criteria were considered, 1162 participants remained in the final study sample (76% men, 24% women, mean age 57 ± 9 years). All the study patients received standard diagnostic and treatment procedures for the secondary prevention of CAD, based on the established guidelines elsewhere [50,51,52,53]. The study and its consent procedures were approved by the Regional Biomedical Research Ethics Committee (project identification code: Kardiogen no. 5, 13 April 2007, No. BE-2-21; 15 January 2009, No. P1-38/2007; 12 September 2009, No. P2-38/2007; 20 April 2010, No. P3-38/2007; 24 October 2012, No. P1-110/2012.) This study is also in accordance with the principles of the Declaration of Helsinki.

### 2.2. Study Procedure

Within two days of admission for CR, all study patients were assessed for (1) socio-demographic information, such as gender, age, and education, and (2) clinical characteristics such as New York Heart Association (NYHA) functional class, history of ACS, and angina pectoris class. Further, usual echocardiography testing to assess the left ventricular ejection fraction was performed on all participants of this study. They also performed EC testing. In order to assess HRQoL, the Short Form (36) Health Survey (SF-36) was used [54]. Patients also independently completed questionnaires, assessing their subjective fatigue level (the Multidimensional Fatigue Inventory, MFI-20) [5,7].

Additionally, depressive and anxiety symptoms (Hospital Anxiety and Depression scale, HADS) [55], state and trait anxiety (The Spielberger State-Trait Anxiety Inventory, STAI) [56] were measured in 200 randomly assigned patients. The number of a sample size was reduced to 200 patients for this analysis involving HADS and STAI due to financial limitations regarding licensing fees of these scales. We employed Lithuanian versions of self-rating scales with good psychometric properties, measured in previous studies [9,57,58,59].

### 2.3. Measures

#### 2.3.1. Multidimensional Fatigue Inventory, MFI-20

Fatigue severity was measured by employing subscales from the MFI-20 original version [5,7]. The MFI, consisting 20 items covers five subscales: (1) General fatigue, (2) physical fatigue, (3) mental fatigue, (4) reduced activity, and (5) reduced motivation. Each domain consists of four items with possible answers on a five-point (1 = “yes, that is true”; 5 = “no, that is not true”) Likert scale. The domain of General fatigue is composed of the general statements about fatigue and reduced functioning, covering physical as well as psychological aspects of fatigue. Physical fatigue concerns physical feelings related to fatigue. Mental fatigue relates to cognitive functioning, such as concentration difficulties. Reduced activity subscale evaluates the impact of psychological and physical factors on the activity level. The lack of motivation to start an activity is reflected by the subscale of reduced motivation. The total score ranges from 4 to 20 on each subscale, and 20 to 100 for total fatigue score with higher score indicating higher fatigue levels.

#### 2.3.2. 36-Item Short Form Medical Outcome Questionnaire, SF-36

The SF-36 is comprised of eight subscales that measure HRQoL on eight domains: (1) Physical and (2) social functioning, (3) role limitations due to physical problems, (4) role limitations due to emotional problems, (5) mental health, (6) energy/vitality, (7) pain, and (8) general health perception. Each of the SF-36 domains is rated on scales from 0 to 100, with a higher total score suggesting better HRQoL [54]. Cronbach α coefficients of all eight domains ranges from 0.56 to 0.85, suggesting adequate internal reliability.

#### 2.3.3. Exercise Capacity Testing, EC

The research cardiologist (J.Br.) measured patients’ EC with a standardized computer-driven bicycle ergometer (Schiller AT-102). Every three minutes the workload was increased by 25 watts (W) [50]. The peak of workload (PW) in watts (W) or MET (1 MET = 3.5 mL of oxygen uptake per kilogram of body weight per minute) at the moment of the termination of the exercise test reflected individual’s EC. Detailed procedures of EC has been reported in our study elsewhere [60].

#### 2.3.4. Hospital Anxiety and Depression Scale, HADS

The self-administered HADS is composed of 14 items, which help to evaluate person’s symptoms of anxiety (HADS-A) and depression (HADS-D) [55]. Total scores on both of the subscales is between 0 and 21. The higher total score suggests the worse symptoms of anxiety and depression [61]. Previous studies suggest good psychometric parameters of the HADS in Lithuanian CAD patients [17,62]. In our study, the internal consistency of the HADS was good (Cronbach α = 0.74; the HADS-A Cronbach α = 0.84).

#### 2.3.5. Spielberger State-Trait Anxiety Inventory, STAI

The STAI was employed to assess state anxiety (STAI-S, 20 items) and trait anxiety (STAI-T, 20 items) [56]. The higher total score on each subscale represents the higher levels of state and trait anxiety. Having good psychometric characteristics in patients with CAD, STAI has been widely used for research purposes in Lithuania [63,64,65]. In our study population, the STAI showed good internal consistency (The STAI-S Cronbach’s α = 0.92; the STAI-T Cronbach’s α = 0.89).

### 2.4. Statistical Analysis

For statistical procedures, we used The SPSS for Windows statistical package (SPSS Inc., Chicago, IL, USA) (version 17.0). The sample size was determined based on the previous reports by Comrey and Lee [66] as well as MacCallum et al. [67] for sample size in factor analysis. The authors [66,67] suggested that in order to assure adequate recovery of population, minimal sampling error, and stable sample factor analysis solutions, as large as the sample size of ≥1000 participants is preferable, even though the 200–500 sample size in some cases might be acceptable.

Descriptive statistics were generated to define our study population with regards to clinical and sociodemographic characteristics. Floor and ceiling effects were used to demonstrate the response distribution and served as a measure of feasibility. If 15% or more respondents achieve the lowest or highest level of the score on a measure, there may be a significant floor or ceiling effect [68]. We further used skewness statistics, kurtosis statistics, and Shapiro–Wilk tests to evaluate the normality of distribution in the variables.

The internal validity of the five MFI subscales has been calculated using corrected-to-total correlations, inter-item correlations, and α-coefficients of Standardized Cronbach. If estimates of a magnitude were higher than 0.7, we considered it as acceptable [69]. The value of the corrected-to-total correlations should be higher than 0.20 and correlations lower than 0.15 are unacceptable [70].

Further, we evaluated convergent validity of the target instrument, while comparing MFI with other related measures administered in the study. Pearson or Spearman correlation coefficients were employed to measure linear associations between the scales of the MFI, the HADS, the STAI, SF-36, and EC measures. Due to financial limitations, to measure the associations between the MFI and the HADS, the STAI, a sample of randomly assigned 200 patients was evaluated.

We used confirmatory factor analysis (CFA) to evaluate whether the original factor structure of the MFI is confirmed for the current sample. The fitness of model with the data was measured by calculating the absolute and comparative fit indices (CFI). Absolute fit indices include chi-square goodness-of-fit (GFI), non-normal fit index (NFI), and root mean square error of approximation (RMSEA). Further analyses resulted in the deletion of 4 items and development of a shortened MFI 16-item version, that helped to improve the factorial structure, while leaving the original number of subscales.

## 3. Results

### 3.1. Sample Characteristics

The final sample was comprised of 1162 CAD patients, 276 women (24%) and 886 men (76%); the average age of participants was 57 ± 9 years. Table 1 represents socio-demographic, clinical, mental distress characteristics, scores of fatigue, and HRQoL of all study patients. Overall, 87% of participants met the criteria for NYHA functional class I-II, while 13% remained within the group of NYHA functional class III. In sum, 37% of study patients had unstable angina pectoris, and 63% were admitted after recent acute MI.

### 3.2. Reliability of MFI 20-Items, MFI-20

The results of three reliability tests of all MFI-20 domains are presented in Table 2. The item redundancy was not detected. Inter-item average of correlations for general fatigue domain was 0.46 (range 0.38–0.58); for physical fatigue domain was 0.53 (range 0.44–0.60); for reduced activity domain was 0.40 (0.30–0.47); for reduced motivation domain was 0.24 (0.15–0.35); for mental fatigue domain was 0.49 (0.35–0.68); and for total score was 0.35 (0.10–0.68). The internal consistency values were reasonable for the MFI-20 total fatigue score and four of the five domains: General fatigue, physical fatigue, reduced activity and mental fatigue Cronbach’s α range: 0.72–0.92. An inadequate value was found for the domain of reduced motivation (Cronbach’s α = 0.55). With regards to the inter-item correlations, a mean value of ≥0.20 was found for the MFI-20 four of the subscales but the lowest correlation was present between items of reduced motivation subscale (0.24, range 0.15–0.35). Item-total correlations were ≥0.20 for all the subscale of MFI-20 (range: 0.28–0.37 to range 0.60–0.68).

### 3.3. Convergent Validity: Relationships of the Subscales of the MFI-20 to Mental Distress Factors, Functional Impairment, and Exercise Capacity

Table 3 displays correlations between MFI-20 total score and fatigue subscales and subscales measuring anxiety and depression symptoms (HADS-A, HADS-D), state and trait anxiety (STAI-S, STAI-T), functional impairment (SF-36) and EC. In overall sample, the strongest correlations were found between the MFI general fatigue (r = −0.51, *p* < 0.001), physical fatigue (r = −0.49, *p* < 0.001), reduced motivation (r = −0.43, *p* < 0.001), total fatigue score (r = 0.551, *p* < 0.001), and the SF-36 subscales measuring energy/vitality. In 200 participants, The MFI reduced motivation (r = 0.50, *p* < 0.001), mental fatigue (r = 0.61, *p* < 0.001), and total fatigue score (r = 0.58, *p* < 0.001) were strongly associated with the STAI Trait anxiety, suggesting excellent convergent validity.

### 3.4. Floor and Ceiling Effects

We explored the possibility for the floor and ceiling effects in all the MFI-20 subscales as well as for total fatigue score. For the general fatigue subscale, *n* = 81 patients (7%) had the lowest possible test score, *n* = 21 patients (2%) had the highest test score. For the physical fatigue subscale, *n* = 72 patients (6%) had the lowest test score, *n* = 54 (5%) the highest test score. For the reduced activity, *n* = 41 patients (3%) had the lowest and *n* = 42 patients (4%) had the highest possible score. For the reduced motivation, *n* = 66 patients (6%) had the lowest test score, while *n* = 4 patients (0.3%) had the highest test score. For the mental fatigue, *n* = 150 patients (13%) had the lowest possible test score, *n* = 17 (1.5%) had the highest possible test score. For the total fatigue score, *n* = 16 patients (1.4%) had the lowest test score, *n* = 1 (0.1%) the highest test score.

### 3.5. Factor Analysis of the MFI-20

We completed a principal components factor analysis using oblique rotation on the MFI-20. The results are presented in Table 4. In summary, the factor analysis solution was complex. In all four factors, the multiple loadings of items had factor-loading values of >0.50. The first factor explained 39% of the variance in the MFI-20, and was comprised of general fatigue and physical fatigue. The second factor was composed of solely all four mental fatigue items that explained 9% of the variance of the MFI-20. However, two of the reduced motivation items and one reduced activity item fell on the third factor, which explaining 6% of the variance of the MFI-20. The fourth factor was loaded by two of the reduced activity items, which explained 5% of the variance in the MFI-20.

Factor 1 (general/physical), factor 2 (mental fatigue), factor 3 (reduced motivation), and factor 4 (reduced activity) were defined by the authors of the original MFI [5]. General fatigue and physical fatigue were two distinct domains on the original MFI instrument. Surprisingly, confirmatory factor analysis testing of the four-factor model showed acceptable fit (CFI = 0.905; GFI = 0.895; NFI = 0.893, RMSEA = 0.077). However, of these 20 items, the four items (3, 9, 17, and 18) were difficult to categorize to one specific factor. In other to enhance the factorial structure of the MFI, a second principal components factor analysis with an oblique rotation was completed on the 16 items.

### 3.6. Factor Analysis of the 16-Item MFI, MFI-16

After removing the four items (3, 9, 17 and 18), the factor analysis with multiple loadings of items had factor-loading values >0.50 across four factors. Detailed results are presented in Table 5. In short, similarly as with previous factor analysis, the first factor was dominated by general fatigue and physical fatigue that was comprised of eight items. The second factor was composed of all four mental fatigue items. Two of the reduced activity items fell under the third factor, while the fourth factor was loaded by two of the reduced motivation items. Of the 16 items of the MFI-16, the first factor explained 39.7% of the variance, while the second, third and fourth factors explained 10.2%, 6.9%, and 6.4% of the variance respectively.

Confirmatory factor analysis testing of the four-factor model from 16 items of the MFI-20 showed acceptable fit (CFI = 0.910; GFI = 0.909; NFI = 0.898, RMSEA = 0.077) (Table 5).

### 3.7. Reliability of MFI-16

The summary of the results of three reliability tests for the all MFI-16 domains is presented in Table 6. Inter-item correlations averaged 0.46 (range 0.38–0.58) for general fatigue, 0.53 (range 0.44–0.60) for physical fatigue, 0.43 for reduced activity, 0.27 for reduced motivation, 0.49 (0.35–0.68) for mental fatigue, and 0.35 (0.12–0.68) for total fatigue score. The internal consistency values were reasonable for the MFI-16 total fatigue score and four of the five subscales: General fatigue, Physical fatigue, Reduced activity, and Mental fatigue Cronbach’s α range: 0.60–0.89. Cronbach’s α = 0.43 was received for the domain of reduced motivation, suggesting inadequate internal reliability. In terms of the inter-item correlations, a mean value of ≥0.20 was calculated for the MFI-16 four of the subscales but the lowest correlation was present between items of reduced motivation subscale (0.24). Item-total correlations were ≥0.20 for all the subscales of MFI-16.

### 3.8. Convergent Validity: Relationships of the Factors of the MFI-16 to Mental Distress Factors, HRQOL, and Exercise Capacity

To evaluate convergent validity in the overall sample, we measured the associations between the MFI-16 and other related constructs. Specifically, Table 7 displays the significant associations between the factors of MFI-16 and subscales measuring anxiety and depressive symptoms (HADS-A, HADS-D), state and trait anxiety (STAI-S, STAI-T), functional impairment (SF-36) and EC. In total sample, the strongest correlations were found between Factor 1 (general/physical fatigue) (r = −0.53, *p* < 0.001), Factor 3 (reduced activity) (r = −0.28, *p* < 0.001) and the SF-36 subscales measuring energy/vitality. In 200 patients, Factor 2 (mental fatigue) (r = 0.61, *p* < 0.001) and Factor 4 (reduced motivation) (r = 0.41, *p* < 0.001) were most strongly correlated with the STAI-T (trait anxiety) and HADS-D (depressive symptoms), suggesting excellent convergent validity. The total fatigue score was most strongly associated with depressive symptoms (r = 0.56, *p* < 0.001) and trait anxiety (r = 0.58, *p* < 0.001).

## 4. Discussion

The purpose of our research was to measure reliability and validity of the MFI as well as to investigate the dimensional structure in the sample of CAD patients after recent ACS. We hypothesized that (1) the internal consistency of a total score and the four MFI domains (i.e., general fatigue, physical fatigue, reduced activity, and mental fatigue) will be satisfactory, and (2) the subjective fatigue (as measured by the MFI) will be linked with mental distress (as measured by HADS and STAI). Both of the hypothesis were met after the completion of the statistical analysis.

The multidimensional structure of the MFI has been found to be comprised of four factors, including general/physical fatigue, mental fatigue, reduced activity, and reduced motivation. The four-factor model used in our study was in line with the original results of Smets et al. [5]. Nevertheless, several issues with the factorial structure were identified. Two items of the original MFI-20 loaded on unexpected dimensions, while the other two did not meet the criteria for factor-loading values >0.50 across the four factors. These inconsistent loading were also reported in previous studies [45,47], suggesting the possibility for further modifications. Respecting the original four factor structure suggested by the MFI-20 authors [5], in the current study, we eliminated the following items (items 3, 9, 17, and 18). After the noted modifications, MFI-16 was an improved version in terms of the factorial structure and confirmatory support was reached, showing acceptable fit. The MFI 16-item four-factor model was further employed to evaluate the internal consistency of all factors.

In terms of factor loadings, even though several previous studies have reported five-factor model of MFI [30,31] or incomplete factorial validity [49], in our study we found that the domains of General fatigue and the domain of Physical fatigue were highly correlated and fell under the same factor. The four-factor model of the MFI appears to be more common in scientific literature and is in accordance with the original study by Smets et al. [5], as well as with other more recent studies [30,37,45,46,71]. It is previously suggested that due to subjective patients’ experience, it might be hard to distinguish between general and physical aspects of fatigue [46]. Nevertheless, the decision whether it is better to merge these two subscales or keep them as separate subscales remained an open question even for the original authors [5]. Based on our study results, the option to merge the General and Physical fatigue subscales of the MFI is recommended when measuring fatigue in those with CAD. However, the five subscales can also be retained, until more data is gathered as suggested by original authors [5].

Furthermore, in our study, we tested the reliability of the MFI-20 and modified version the MFI-16 while using inter-item correlations, corrected item-to-total correlations and Cronbach’s α values. Inter-item correlations and corrected-to-total correlations suggested adequate reliability. All factors showed moderate to acceptable [69] Cronbach α coefficients ranging from 0.60 to 0.82, except for reduced motivation factor (Cronbach α = 0.43). The lower Cronbach α coefficient for reduced motivation factor is consistent with previous findings [45,46,47,48], thus the results of a current study might not be caused due to cultural differences but rather reflects the psychometric properties of the MFI-20. Nevertheless, further investigation for reduced motivation subscale is necessary to address this issue. Overall, the current study suggests adequate reliability for the MFI, except for one subscale of reduced motivation.

Further, the current study has also shown that convergent validity of Lithuanian MFI-20 as well as modified MFI-16 is good: Each subscale and factor was correlated with closely related constructs of mental distress (anxiety and depressive symptoms (HADS-A, HADS-D), state and trait anxiety (STAI-S, STAI-T)), HRQoL as presented by all SF-36 subscales, as well as the level of objective EC (all *p*’s ≤ 0.001). Our findings are in line with the previous research by Smets et al. [72] Fillion et al. [45], Lin et al. [31] in terms of showing high correlations with previously mentioned mental distress and HRQoL scales.

Additionally, we have also investigated floor and ceiling effect for the MFI. Our findings yielded plausible results as no ceiling or floor effects were detected. Similarly, in a recent study by Antonio et al. [41] it was found that 2.5% of the CAD patients reported lowest point on total fatigue score, while none has scored the highest score. In our study we expanded the knowledge and further explored not only floor and ceiling affects for a total fatigue score but also for separate domains, where participants number with the lowest scores ranged from 3 to 13% and with the highest scores ranged from 0.3 to 5%. Thus, in none of the domains participants number topped 15% indicating a significant floor or ceiling effect [68].

Several ideas may explain the inconsistent factorial structure of the MFI-20 items and the need to eliminate four items in order to improve the dimensional structure. First, all our patients had been in the CR for only two weeks, thus questions like “I get little done“ (question 17) or “I feel very active“ (question 3) might have been answered based on the changed context of physical settings, rather than internal state and health condition. Secondly, due to cultural differences and subjective understanding of fatiguing experience, questions like “I dread having do things“ (question 9) or “I don‘t feel like doing anything“ might have been attributed to somewhat different concept than reduced motivation, possibly to symptomatology of CAD.

From occupational health perspective, up to 91% of the CAD patients after CR [73] returns to the workplace, while successful vocational reintegration is essential to those individuals and their psychosocial functioning [74]. Up till now, occupational therapists have been using various instruments, including those that assess anxiety and depressive symptoms as well as health related outcome measures of former CAD patients [74]. The MFI could serve as a useful instrument for evaluating and monitoring different fatigue characteristics in individuals with CAD that could further assist in tailoring individualized work conditions, workload or daily schedule that may help them to better adapt to their health changes at work.

### 4.1. Study Limitations

Despite of our consistent findings, several limitations should be noted. First, test-retest reliability was not evaluated, precluding us from making interpretation on replicability of the given results more than once in the same population. Secondly, the study was completed in a large cohort of Lithuanian CAD patients, thus generalizability for other cultural cohorts should be assumed and interpreted with caution. Finally, even though we achieved the original four factor structure, to meet this assumption, the adjustment had to be completed and the original 20-item MFI was reduced to 16-items. Despite the potential limitation, our study included a large sample size of CAD patients, suggesting high confidence of generalizability of the results in Lithuanian patients with CAD.

### 4.2. Future Directions

Even though the MFI is a commonly used instrument, its structure still struggles to find perfect suitability among the various diseases and cultural adaptations. In order to address these issues, we developed a shorter MFI version of 16 items with good factorial structure and sound psychometric qualities. Nevertheless, further studies in various clinical samples are warranted to further address the issues related to the MFI psychometric properties, especially factorial structure, and internal consistency of reduced motivation subscale. Considering the validity of the MFI, the improvements in the future studies can be made in terms of random selections of the CAD patients as well as studies in more diverse cultural cohorts, allowing the clinicians and researchers in related fields to draw more confident conclusions for the generalizability of the study results.

## 5. Conclusions

This is the first study showing the MFI as an adequate instrument to evaluate the level of fatigue in Lithuanian clinical settings. Nevertheless, the modified MFI version of MFI-16 can be considered as psychometrically improved instrument to measure the multidimensional construct of fatigue, as it was found to have a sound and complete factorial structure in CAD patients. MFI-16 may assist in early detection and characterization of fatigue in cardiac patients undergoing cardiovascular rehabilitation after recent acute coronary event. This may help to tailor individualized treatment program and to not only reduce the levels of fatigue but also positively affect patients’ health related quality of life and emotional state, including depressive and anxiety symptoms.

This latest questionnaire is tailored to CAD patients, and clinicians should find it useful to test the fatigue of these patients. However, the subscale of reduced motivation should be considered with caution due to possible threat to internal consistency.

## Figures and Tables

**Table 1 ijerph-17-08003-t001:** Sociodemographic and clinical characteristics of all patients.

Characteristic	N = 1162	
	Mean	SD
Age	57.34	9.09
	**N**	**Percent**
Gender:		
Male	886	76.2
Female	276	23.8
Education:		
Up to 8 years	86	7.4
High school graduate	577	49.7
College/university degree	499	42.9
Diagnosis:		
Unstable angina pectoris	430	37.0
Acute myocardial infarction	732	63.0
NYHA class:		
I	86	7.4
II	921	79.3
III	155	13.3
HF class:		
A	111	9.6
B	817	70.3
C	234	20.1
Arterial hypertension	951	81.8
Left ventricular ejection fraction ≤40%	115	9.9
	**Mean**	**SD**
Left ventricular ejection fraction	51.35	8.45
Exercise capacity workload (W)	72.90	26.95
	**N**	**Percent**
Medication Use:		
Nitrates	267	23.0
Beta-blockers	1027	88.4
ACE inhibitors	941	81.0
Diuretics	169	14.5
Benzodiazepines	162	13.9
	**Mean**	**SD**
State Trait Anxiety Inventory:		
State anxiety	37.27	10.51
Trait anxiety	42.8	9.52
	**N**	**Percent**
Anxiety symptoms (HADS-A):		
Total score <8	140	70.0
Total score ≥8	60	30.0
Depressive symptoms (HADS-D):		
Total score <8	175	87.5
Total score ≥8	25	12.5
	**Mean**	**SD**
MFI-20 score		
General fatigue	10.76	3.97
Physical fatigue	11.75	4.34
Reduced activity	12.33	3.90
Reduced motivation	9.91	3.43
Mental fatigue	9.77	4.02
Total Fatigue Score	54.52	16.18
SF-36		
Physical functioning	68.98	19.73
Role limitation due to physical problems	30.16	37.27
Role limitation due to emotional problems	52.70	43.72
Social functioning	66.84	26.24
Mental health	68.28	19.11
Energy/vitality	59.09	20.70
Pain	51.04	27.52
General health perception	53.15	18.86

HADS-A—Anxiety subscale of the Hospital Anxiety and Depression Scale; HADS-D—Depression subscale of the Hospital Anxiety and Depression Scale; NYHA—New York Heart Association; SF-36—Medical Outcomes Study 36-Item Short Form Health Survey; MFI-20—Multidimensional Fatigue Inventory; W—Watts; HF—heart failure.

**Table 2 ijerph-17-08003-t002:** MFI-20 scale item characteristics and internal consistency reliabilities.

Fatigue Characteristics	Mean	SD	Inter-Item Correlation		Corrected-to-Total Correlation	Coefficient α (If Item Deleted)	Standardized Cronbach’s α
			Mean	Range	Range	Range	
General Fatigue	10.76	3.97	0.46	(0.38–0.58)	0.52–0.65	0.67–0.75	0.77
Physical Fatigue	11.75	4.34	0.53	(0.44–0.60)	0.60–0.68	0.76–0.79	0.82
Reduced Activity	12.33	3.90	0.40	(0.30–0.47)	0.47–0.57	0.63–0.69	0.72
Reduced Motivation	9.91	3.43	0.24	(0.15–0.35)	0.28–0.37	0.45–0.52	0.55
Mental Fatigue	9.77	4.02	0.49	(0.35–0.68)	0.46–0.70	0.69–0.73	0.79
Total Fatigue Score	54.52	16.1	0.35	(0.10–0.68)	0.32–0.72	0.91–0.92	0.92

MFI-20—Multidimensional Fatigue Inventory.

**Table 3 ijerph-17-08003-t003:** Convergent Validity: Pearson Correlation Coefficients between the MFI-20, the HADS, the STAI, the SF-36, and EC testing in overall sample.

Clinical Characteristics	MFI-20					Total Fatigue Score
	General Fatigue	Physical Fatigue	Reduced Activity	Reduced Motivation	Mental Fatigue	
	r (*p* < 0.001)					
HADS						
Anxiety symptoms	0.416	0.363	0.323	0.319	0.451	0.466
Depressive symptoms	0.551	0.509	0.484	0.525	0.533	0.563
STAI						
State anxiety	0.587	0.499	0.434	0.507	0.567	0.541
Trait anxiety	0.553	0.446	0.424	0.496	0.613	0.582
SF-36						
Physical functioning	−0.459	−0.446	−0.386	−0.355	−0.299	−0.475
Role limitation due to physical problems	−0.290	−0.304	−0.273	−0.229	−0.172	−0.310
Role limitation due to emotional problem	−0.278	−0.248	−0.243	−0.257	−0.269	−0.315
Social functioning	−0.376	−0.341	−0.300	−0.288	−0.296	−0.391
Mental health	−0.418	−0.365	−0.323	−0.360	−0.426	−0.461
Energy/vitality	−0.514	−0.494	−0.434	−0.405	−0.412	−0.551
Bodily pain	−0.295	−0.294	−0.241	−0.207	−0.182	−0.298
General health	−0.484	−0.486	−0.401	−0.410	−0.344	−0.518
Exercise capacity	−0.307	−0.316	−0.279	−0.317	−0.193	−0.331

MFI-20—Multidimensional Fatigue Inventory; HADS—the Hospital Anxiety and Depression Scale; STAI—State Trait Anxiety Inventory; SF-36—Medical Outcomes Study 36-Item Short Form Health Survey.

**Table 4 ijerph-17-08003-t004:** Factor analysis of the Multidimensional Fatigue Inventory (MFI) 20-item responses.

Fatigue Characteristics and Items	Factors			
	1	2	3	4
General Fatigue				
I feel fit	0.529			
I feel tired	0.726			
I feel rested	0.526			
I tired easily	0.723			
Physical Fatigue				
Physically I feel I am in a bad condition	0.642			
Physically I feel I am in an excellent condition	0.619			
Physically I feel I am in a bad condition	0.737			
Physically I can take on a lot	0.658			
Reduced Activity				
I feel very active			0.609	
I think I do a lot in a day				0.658
I think I do very little in a day				0.774
I get little done	0.607			
Reduced Motivation				
I feel like doing all sorts of nice things			0.630	
I dread having to do things				
I have a lot of plans			0.594	
I don’t feel like doing anything				
Mental Fatigue				
When I am doing something, I can keep my thoughts on it		0.706		
I can concentrate well		0.750		
It takes a lot of effort to concentrate on things		0.772		
My thoughts easily wander		0.626		
Alpha	0.90	0.79	0.66	0.60

Note: Extraction Method: Principal Component Analysis. Rotation Method: Varimax with Kaiser Normalization. Factor loadings less than 0.5 were not listed in the table. Kaiser–Meyer–Olkin index = 0.937, *p*-value for Bartlett’s Test of Sphericity is less than 0.001. Cumulative percent of explained variance is 59.3%.

**Table 5 ijerph-17-08003-t005:** Factor analysis of Multidimensional Fatigue Inventory (MFI) 16-items responses.

Fatigue Characteristics and Items	Factors			
	1	2	3	4
General Fatigue				
I feel fit	0.588			
I feel tired	0.739			
I feel rested	0.579			
I tired easily	0.739			
Physical Fatigue				
Physically I feel I am in a bad condition	0.660			
Physically I feel I am in an excellent condition	0.673			
Physically I feel I am in a bad condition	0.757			
Physically I can take on a lot	0.716			
Reduced Activity				
I think I do a lot in a day			0.745	
I think I do very little in a day			0.796	
Reduced Motivation				
I feel like doing all sorts of nice things				0.555
I have a lot of plans				0.732
Mental Fatigue				
When I am doing something, I can keep my thoughts on it		0.736		
I can concentrate well		0.790		
It takes a lot of effort to concentrate on things		0.772		
My thoughts easily wander		0.638		
Alpha	0.89	0.79	0.60	0.43

Note: Extraction Method: Principal Component Analysis; Rotation Method: Varimax with Kaiser Normalization. Factor loadings less than 0.5 were not listed in the table. Kaiser–Meyer–Olkin index = 0.919, *p*-value for Bartlett’s Test of Sphericity is less than 0.001. Cumulative percent of explained variance is 63.2%.

**Table 6 ijerph-17-08003-t006:** Multidimensional Fatigue Inventory (MFI) 16-items characteristics and internal consistency reliabilities.

Fatigue Characteristics	Mean	SD	Inter-Item Correlation		Corrected-to-Total Correlation	Coefficient α (If Item Deleted)	Standardized Cronbach’s α
			Mean	Range	Range	Range	
General Fatigue	10.76	3.97	0.46	0.38–0.58	0.52–0.65	0.67–0.75	0.77
Physical Fatigue	11.75	4.34	0.53	0.44–0.60	0.60–0.68	0.76–0.79	0.82
Reduced Activity	6.45	2.26	0.43	-	0.43	-	0.60
Reduced Motivation	5.26	2.07	0.27	-	0.27	-	0.43
Mental Fatigue	9.77	4.02	0.49	0.35–0.68	0.46–0.70	0.69-0.73	0.79
Total Fatigue Score	43.99	13.03	0.35	0.12–0.68	0.14–0.58	0.83-0.90	0.89

**Table 7 ijerph-17-08003-t007:** Convergent validity: Pearson Correlation Coefficients between the factors of Multidimensional Fatigue Inventory (MFI) 16—items, the HADS, the STAI, the SF-36, and exercise capacity in the overall sample.

Clinical Characteristics	Factors				Total Fatigue Score
	1	2	3	4	
	r (*p* < 0.001)				
HADS					
Anxiety	0.410	0.451	0.304	0.177	0.461
Depressive symptoms	0.558	0.533	0.374	0.424	0.561
STAI					
State anxiety	0.570	0.567	0.342	0.381	0.539
Trait anxiety	0.524	0.613	0.333	0.407	0.576
SF-36					
Physical functioning	−0.478	−0.299	−0.237	−0.244	−0.461
Role limitation due to physical problems	−0.315	−0.172	−0.180	−0.155	−0.299
Role limitation due to emotional problem	−0.277	−0.269	−0.161	−0.150	−0.302
Social functioning	−0.378	−0.296	−0.197	−0.216	−0.388
Mental health	−0.413	−0.426	−0.217	−0.246	−0.458
Energy/vitality	−0.532	−0.412	−0.282	−0.326	−0.549
Bodily pain	−0.311	−0.182	−0.180	−0.121	−0.294
General health	−0.512	−0.344	−0.264	−0.329	−0.514
Exercise capacity	−0.329	−0.193	−0.185	−0.232	−0.315

HADS—the Hospital Anxiety and Depression Scale; STAI—State Trait Anxiety Inventory; SF-36—Medical Outcomes Study 36-Item Short Form Health Survey; Factor 1—General/Physical fatigue; Factor 2—Mental fatigue; Factor 3—Reduced activity; Factor 4—Reduced motivation.

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
