# Peer review of "Validation of the Multidimensional Fatigue Inventory with Coronary Artery Disease Patients"

_ijerph, 2020, doi:10.3390/ijerph17218003_

Round 1

Reviewer 1 Report

This paper presents a study to evaluate the Multidimensional Fatigue Inventory in patients with coronary artery disease (CAD).

It is clear that the fatigue is a very important issue that deserves to be studied, especially in in patients with coronary artery disease, but I'm afraid there are some points in the paper which could deserve further explanations.

  1. Please describe the hypothesis in this section.
  2. Study participants. How was the sample size calculated?
  3. What is the study design? When was it done?
  4. Discussion section. I recommend adding a section with the limitations of the study.

Also, other similar current studies are not reflected. Could you expand the comparison with other studies?

I have found studies published in recent years on arteriosclerosis, validation, reliability, blood pressure. Do you think it could be related to your study?

Update the introduction or discussion and include it in your references, if necessary.

https://pubmed.ncbi.nlm.nih.gov/30361193/

https://pubmed.ncbi.nlm.nih.gov/29246880/

  1. Could you suggest any future lines to expand the internal and external validity of this research?

Reviewer 2 Report

This study measured the reliability and validity of the MFI in a cohort of Lithuanian CAD patients after recent ACS. The study found adequate reliability except for the sub scale of reduced motivation.

The grammar throughout the paper requires some attention, for example in line 32: as "a" subjective experience

Why were 1287 patients chosen for this study? Is there a power calculation?

During which months/years was the study conducted?

Why were depressive and anxiety symptoms only measured in 200 individuals? Why not the entire population?

Please provide more information not he psychometric properties of the MFI-20

Do you believe that these results would be replicated in different cultural cohorts?

Reviewer 3 Report

The manuscript by Stonciene et al has shown factorial structure and satisfactory psychometric characteristics of coronary artery disease patients in Lithuania. Tha manuscript is well written and methodologies are very sound. I have one major concern that:

The authors have not well described the findings they just put the previous publications as reference and told the data is in line. This seems like weak representation of results.

I will ask authors to provide to summarize the main findings in bulleted format and put in a new section as clinical perspectives. 

Author also need to mention importance of the study in introduction section and in discussion they need to mention how these observations are going to help to solve mental fatigue in CAD patients.

Round 2

Reviewer 1 Report

This paper has been suitably modified and improved. Therefore, the manuscript is accepted for publication.

Author Response

We are grateful for all your efforts and time given for the review process of our manuscript.